Citation: *Molecular Systems Biology* 9:665
www.molecularsystemsbiology.com

# Nucleotide degradation and ribose salvage in yeast

Yi-Fan Xu[1,2], Fabien Létisse[3], Farnaz Absalan[4], Wenyun Lu[1], Ekaterina Kuznetsova[5], Greg Brown[5], Amy A Caudy[6], Alexander F Yakunin[5], James R Broach[4] and Joshua D Rabinowitz[1,2,*]

[1] Lewis Sigler Institute for Integrative Genomics, Princeton University, Princeton, NJ, USA, [2] Department of Chemistry, Princeton University, Princeton, NJ, USA, [3] Université de Toulouse, INSA, UPS, INP; LISBP, Toulouse, France, [4] Department of Molecular Biology, Princeton University, Princeton, NJ, USA, [5] Department of Chemical Engineering and Applied Chemistry, Banting and Best Department of Medical Research, University of Toronto, Toronto, ON, Canada and [6] Donnelly Centre for Cellular and Biomolecular Research, University of Toronto, Toronto, Canada
* Corresponding author. Chemistry and Genomics, Princeton University, 241 Carl Icahn Laboratory, Princeton, NJ 08540, USA. Tel.: + 1 609 258 8985; Fax: + 1 609 258 3565; E-mail: joshr@princeton.edu

Nucleotide degradation is a universal metabolic capability. Here we combine metabolomics, genetics and biochemistry to characterize the yeast pathway. Nutrient starvation, via PKA, AMPK/SNF1, and TOR, triggers autophagic breakdown of ribosomes into nucleotides. A protein not previously associated with nucleotide degradation, Phm8, converts nucleotide monophosphates into nucleosides. Downstream steps, which involve the purine nucleoside phosphorylase, Pnp1, and pyrimidine nucleoside hydrolase, Urh1, funnel ribose into the nonoxidative pentose phosphate pathway. During carbon starvation, the ribose-derived carbon accumulates as sedoheptulose-7-phosphate, whose consumption by transaldolase is impaired due to depletion of transaldolase's other substrate, glyceraldehyde-3-phosphate. Oxidative stress increases glyceraldehyde-3-phosphate, resulting in rapid consumption of sedoheptulose-7-phosphate to make NADPH for antioxidant defense. Ablation of Phm8 or double deletion of Pnp1 and Urh1 prevent effective nucleotide salvage, resulting in metabolite depletion and impaired survival of starving yeast. Thus, ribose salvage provides means of surviving nutrient starvation and oxidative stress.
*Molecular Systems Biology* **9**: 665; published online 14 May 2013; doi:10.1038/msb.2013.21
*Subject Categories:* RNA; cellular metabolism
*Keywords:* autophagy; mass spectrometry; metabolism; nutrient starvation; *Saccharomyces cerevisiae*

## Introduction

In natural environments, microbes face myriad environmental stressors including starvation for essential nutrients, such as carbon, nitrogen and phosphorous. Yeast has evolved metabolic responses to these stressors, which enable their survival and effective resumption of growth when conditions improve. Recent evidence suggests that these metabolic responses are substantially hardwired in metabolic enzymes themselves (Xu *et al*, 2012a, b; Link *et al*, 2013). In parallel, with this intrinsic adaptation of metabolic network, proteins such as kinases sense metabolic conditions and activate signaling cascades to coordinate metabolism and overall cellular activity. Important examples of such signaling enzymes include protein kinase A (PKA, activated by glucose), AMP-activated protein kinase (SNF1, repressed by glucose), and target of rapamycin (TOR, activated by abundant nitrogen) (Komeili *et al*, 2000; Wilson and Roach, 2002; Zaman *et al*, 2009; Ratnakumar and Young, 2010).

A particularly important function of these kinases is to control levels of ribosomes, which constitute about 10% of yeast dry weight. This is achieved through regulation both of ribosome production and of ribosome degradation via

autophagy (ribophagy) (Ertugay and Hamamci, 1997; Martin *et al*, 2004; Lee *et al*, 2007; Yorimitsu *et al*, 2007; von der Haar, 2008; Stephan *et al*, 2009; Oliveira and Sauer, 2012). Ribosomes are composed of roughly equal amounts of protein and RNA (Ben-Shem *et al*, 2011). Thus, their degradation yields both amino acids and nucleotides. Catabolism of nucleotides involves their dephosphorylation into nucleosides, and subsequent hydrolysis into ribose and bases (Tozzi *et al*, 2006). While many hydrolytic enzymes with nucleotidase and nucleosidase activities exist in various organisms (Lecoq *et al*, 2001; Kurtz *et al*, 2002; Mitterbauer *et al*, 2002; Nakanishi and Sekimizu, 2002; Canduri *et al*, 2004; Hunsucker *et al*, 2005), the ones that are actually responsible for nucleotide degradation in cells have remained undefined.

The carbonaceous product of nucleotide degradation is ribose-5-phosphate, which must be converted into glycolytic intermediates in order to extract energy and generate anabolic precursors. Such conversion is carried out by the nonoxidative pentose phosphate pathway, a series of five reversible reactions whose net stoichiometry is 3 ribose-5-phosphate $\leftarrow\rightarrow$ 2 fructose-6-phosphate (F6P) + 1 glyceraldehyde-3-phosphate (Figure 1). Both F6P and glyceraldehyde-3-phosphate can be catabolized by glycolysis to yield NADH and ATP.

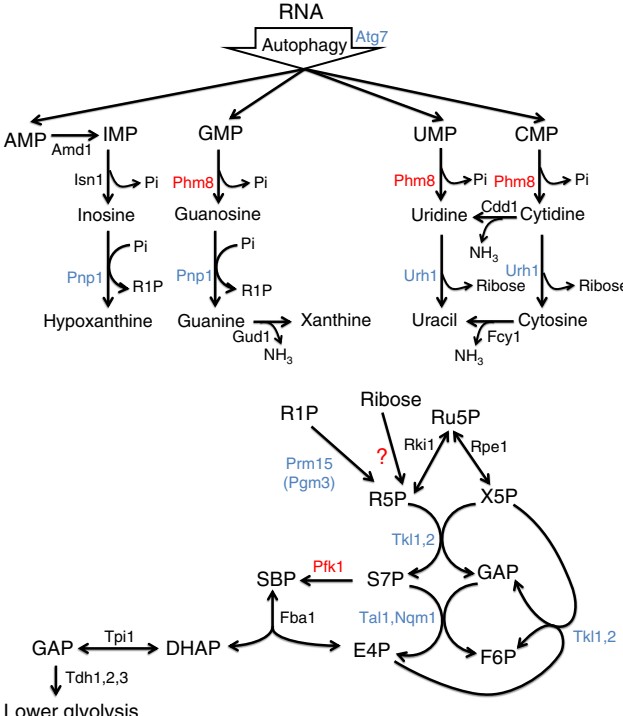

**Figure 1** Yeast pathway map of nucleotide degradation and ribose salvage. The pathway begins with the degradation of RNA. In starvation, this is triggered by macroautophagy (ribophagy). The resulting nucleotides are degraded into nucleosides except for AMP, which is first converted to IMP (Walther *et al*, 2010). Purine nucleosides are then converted into ribose-1-phosphate and base, whereas pyrimidines are hydrolyzed into ribose and base. Both ribose and ribose-1-phosphate are converted into ribose-5-phosphate, which enters the nonoxidative branch of PPP. Gene annotations that are taken from definitive prior literature are shown in black; those confirmed by experiments in this study are shown in blue, and those newly assigned in this study are shown in red. Incorrect and missing annotations in the current genome scale metabolic model (Herrgard *et al*, 2008) (http://www.comp-sys-bio.org/yeastnet/) are listed in Supplementary Table S1. The enzyme converting ribose to R5P remains unclear. G6P, glucose-6-phosphate; FBP, fructose-1,6-bisphosphate; GAP, glyceraldehyde-3-phosphate; DHAP, dihydroxyacetone phosphate; R5P, ribose-5-phosphate; X5P, xylulose-5-phosphate; Ru5P, ribulose-5-phosphate; E4P, erythrose-4-phosphate.

In addition, F6P can isomerize into glucose-6-phosphate, which in turn can be catabolized by the oxidative pentose phosphate pathway (PPP) to generate NADPH, a key high energy electron donor required for reductive biosynthesis and redox defense (Grant, 2001; Ralser *et al*, 2007).

A central intermediate in the nonoxidative PPP is sedoheptulose-7-phosphate. When net flux is from ribose-5-phosphate towards glycolytic intermediates, S7P is produced by the enzyme transketolase and consumed by transaldolase. It was previously shown that, in the absence of transaldolase, xylose-grown *E. coli* could also utilize the glycolytic enzyme phosphofructokinase (Pfk1) to phosphorylate S7P into SBP, which is further converted to erythrose-4-phosphate (E4P) and glycolytic intermediate dihydroxyacetone phosphate (Nakahigashi *et al*, 2009). No such reaction has been reported for wild-type cells or other organisms, although it has recently been found that SBP can be produced in wild-type yeast from E4P and dihydroxyacetone phosphate and subsequently

dephosphorylated by the phosphatase Shb17 to yield S7P and thereby drive ribose production (Clasquin *et al*, 2011).

Here we use a combination of metabolomics, genetics and biochemistry to map the nucleotide degradation pathway in yeast (Figure 1), and to explore its regulation and physiological function. We show that during nutrient starvation, yeast accumulate both nucleosides and bases, indicating active nucleotide degradation. Surprisingly, during carbon starvation, they also accumulate S7P and SBP. By tracking the impact of gene deletions on the levels of these metabolites, we identify Phm8, which is currently annotated as a lysophosphatidic acid phosphatase (Reddy *et al*, 2008), as the main yeast nucleotidase. In addition, we determine the cause and physiological function of the S7P accumulation. We show that S7P accumulation and more generally nucleotide degradation enable metabolic homeostasis and survival of starving yeast.

## Results

### Starvation triggers degradation of RNA into nucleosides, bases, and sedoheptulose-7-phosphate

To characterize the nucleotide degradation pathway, we first identified physiological conditions that robustly induced its activity. To evaluate whether nutrient starvation would do so, we used liquid chromatography—mass spectrometry (LC-MS) to measure the changes in the metabolome profile of *S. cerevisiae* cells subjected to abrupt transition from complete minimal medium to media lacking carbon, nitrogen or phosphate. Nucleotide monophosphates (NMPs), the direct product of RNA degradation, increased only slightly. Nucleosides and bases, however, rapidly accumulated to greater than 10-fold basal levels (Figure 2). Moreover, while sugar phosphates in upper glycolysis decreased by more than 10-fold upon carbon starvation, PPP intermediates sedoheptulose-7-phosphate (S7P) and sedoheptulose-1,7-bisphosphate (SBP) accumulated, with S7P becoming the most abundant sugar phosphate in the cell (concentration ~5 mM).

To explore the origin of the accumulating metabolites, we conducted a metabolomic experiment involving feeding of U-[13]C-glucose for 70 min followed by carbon starvation (Figure 3A). This regimen yields complete labeling of metabolic intermediates, including free nucleotides, but only partial labeling of macromolecule stores (protein, glycogen, RNA and DNA). Such transient labeling allowed us to determine the extent to which metabolites arise from macromolecule breakdown (Yuan and Rabinowitz, 2007). We found that the proportion of unlabeled nucleosides, bases and sedoheptulose species increased over time after carbon starvation (Figure 3B, closed symbols), indicating that these metabolites were predominantly derived from degradation of macromolecules. Consistent with those macromolecules being RNA, the total cellular RNA concentration decreased over the same time period (Supplementary Figure S1A).

A likely mechanism of RNA degradation is via macroautophagy. We accordingly measured metabolite concentrations in the *atg7* deletion strain, which fails to elicit macroautophagy upon starvation. Consistent with a requirement for macroautophagy in RNA catabolism, during carbon starvation the

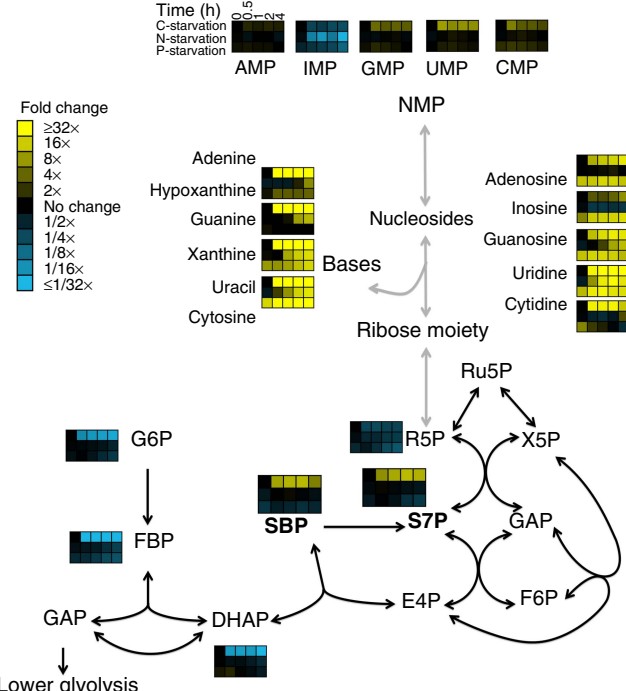

**Figure 2** Starvation triggers accumulation of nucleosides, bases and S7P. Wild-type *S. cerevisiae* cells growing in minimal media were switched to minimal media containing no carbon, no nitrogen or no phosphate. After the indicated duration of starvation, the metabolome was quantified by LC-MS. For media composition, see Supplementary Table S2. For the list of absolute concentration of metabolties, see Supplementary Table S3. Data are shown in heat map format, with each line reflecting the dynamics of a particular compound in a particular culture condition. Metabolite levels of biological duplicates were averaged, normalized to cells growing steadily in glucose (time zero), and the resulting fold changes log$_2$ transformed.

*atg7* deletion strain did not accumulate nucleosides, bases and sedoheptulose species, as normally observed in a wild-type strain (Figure 3C). Moreover, the deletion strain showed less increase in unlabeled species when transient feeding of $^{13}$C-labeled glucose was followed by carbon starvation (Figure 3B, open symbols). Similarly, the increase in nucleosides and bases during nitrogen starvation was absent in the *atg7* deletion strain (Supplementary Figure S1B). In contrast, their accumulation during phosphate starvation was not impaired by *atg7* deletion, indicating that phosphate starvation activates RNA degradation via an alternative mechanism. Nonetheless, our results demonstrate that RNA degradation upon carbon and nitrogen starvation requires macroautophagy.

We examined whether the mechanism by which nutrient starvation triggers autophagy and subsequent nucleoside and base accumulation are through canonical nutrient-sensing kinases, such as cyclic-AMP dependent protein kinase (PKA), AMP-activated protein kinase (SNF1), and target of rapamycin complex 1 (TORC1). PKA is an inhibitor of macroautophagy but is inactivated upon glucose removal (Yorimitsu *et al*, 2007; Stephan *et al*, 2009). SNF1 is active in the absence of glucose and is required for growth on most carbon sources other than glucose (Zaman *et al*, 2008; Broach, 2012). To test if ribophagy requires inactivation of PKA and/or activation of SNF1, we

performed carbon starvation on a *bcy1* deletion strain, which exhibits constitutively activated PKA activity, and on a *snf1* deletion strain, which lacks SNF1 activity. Both strains had substantially decreased accumulation of nucleosides, bases, and sedoheptulose species (Figure 3D). Thus, both activation of SNF1 and inactivation of PKA are required for robust induction of autophagy in response to carbon starvation. TORC1 is an inhibitor of autophagy, which is inactivated upon nitrogen removal (Noda and Ohsumi, 1998). To test if TORC1 inactivation is sufficient to initiate RNA degradation, we treated wild-type cells growing in the presence of nitrogen with rapamycin and analyzed metabolites over time following treatment. Rapamycin treatment resulted in accumulation of nucleosides and bases to an extent even higher than those observed following nitrogen starvation (Figure 3E). Thus, TORC1 inactivation is *per se* sufficient to activate autophagy.

## Phm8 is the physiological yeast nucleotidase

The significant accumulation of nucleosides upon nutrient starvation or rapamycin addition indicates not only that starvation induces RNA degradation but also that the resulting nucleotides are dephosphorylated to nucleosides. The enzyme catalyzing the dephosphorylation step under physiological conditions has not been identified in any organism. Sdt1 has been proposed as the responsible enzyme in yeast, since Sdt1 dephoshorylates UMP and CMP into uridine and cytidine rapidly *in vitro* (Nakanishi and Sekimizu, 2002). Contrary to this expectation, our metabolomic analysis of an *sdt1* strain subjected to carbon starvation did not reveal altered nucleo-tide salvage pathway metabolite concentrations (Figure 4A). The enzyme Phm8 is homologous to Sdt1 (42% amino-acid sequence identity). Phm8 has been proposed to be a cytosolic lysophosphatidic acid (LPA) phosphatase involved in LPA hydrolysis in response to phosphate starvation (Reddy *et al*, 2008). Over-expression of Phm8 decreased LPA in yeast but deletion of *PHM8* did not markedly increase LPA levels, prompting the hypothesis that Phm8 is a minor LPA phosphatase. Interestingly, PHM8, but not SDT1, is expressed under various stress conditions including carbon, nitrogen and phosphate starvation (Gasch *et al*, 2000; Bradley *et al*, 2009; Klosinska *et al*, 2011) (Figure 4B). These expression data together with *PHM8*'s sequence similarity to *SDT1* suggested to us that it might serve as a nucleotidase in the ribose salvage pathway.

We measured the enzymatic activity of purified Phm8 against 90 different phosphorylated compounds. Of these, the preferred substrates were nucleotide monophosphates (specific activity > 500 nmol/mg/min for AMP, GMP, CMP, and UMP) (Figure 4C). In contrast, the activity toward LPA was only ~2 nmol/mg/min (Reddy *et al*, 2008). Thus, Phm8's primary enzymatic activity is as a nucleotidase. Kinetic characterization of Phm8 and Sdt1 toward their most preferred substrate CMP shows that Phm8 has higher $k_{cat}$ and lower $K_m$ (Figure 4D); thus, Phm8 is more active than Sdt1. To evaluate Phm8's physiological role, we measured metabolite levels in the *phm8* strain upon carbon starvation. Although nucleosides and bases still increased, *phm8* deletion blocked the increase in S7P and SBP levels (Figure 4A). Thus, while other nucleotidases can apparently partially substitute for Phm8,

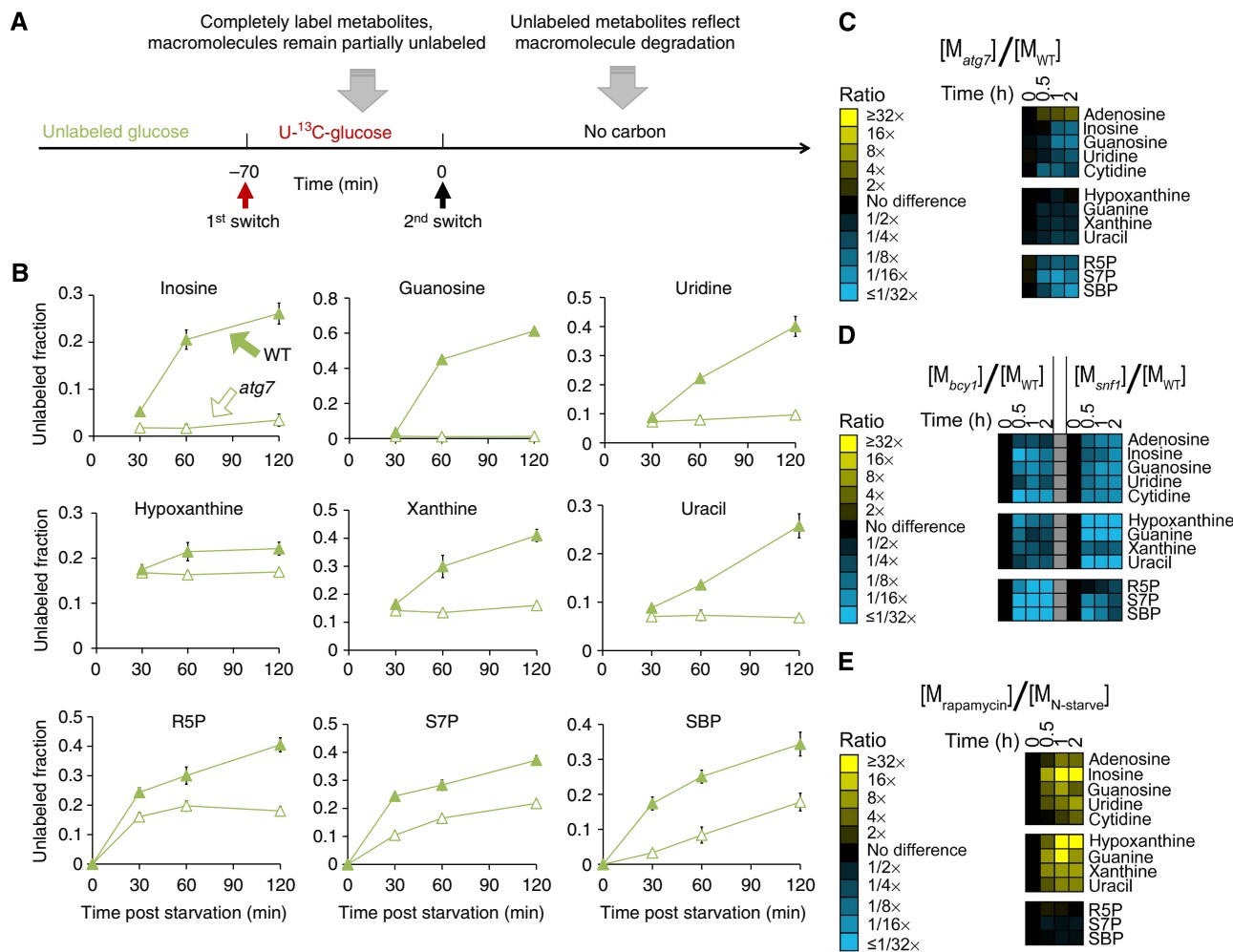

**Figure 3** In carbon starvation, nucleosides and PPP compounds are produced by RNA degradation via autophagy in response to kinase signals. (**A**) Experimental design for demonstrating metabolite production via macromolecule degradation. Yeast cells were grown on 2% unlabeled glucose, and then switched to U-$^{13}$C-glucose for 70 min, which completely labels free metabolite but only partially labels macromolecules. Thereafter, glucose was removed and the metabolome analyzed by LC-MS. (**B**) Fraction of unlabeled nucleosides, nucleic bases and PPP intermediates as a function of starvation time, in wild-type and autophagy deficient (*atg7* deletion) yeast. The *x* axis represents minutes after carbon starvation, and the *y* axis represents fraction of unlabeled metabolites (mean ± range of *N* = 2 biological replicates). (**C**) Ratio of metabolite levels in *atg7* strain versus wild-type strain in carbon starvation. (**D**) Ratio of metabolite levels in *bcy1* strain and *snf1* strain versus wild-type strain in carbon starvation. (**E**) Ratio of metabolite levels in rapamycin treatment versus nitrogen starvation for wild-type yeast. In (**C**) to (**E**), all reported values are log$_2$ transformed ratios; data are mean of duplicate samples at each time point.

Phm8 is required to maintain sufficient nucleotide catabolic flux to elevate S7P and SBP.

## Nucleosides are converted into ribose-5-phosphate by Pnp1, Urh1, and Pgm3

Nucleosides are cleaved into nucleic bases and ribose moieties via either a phosphorylase, which yields ribose-1-phosphate, or a hydrolase, which yields unphosphorylated ribose. In yeast, purine nucleoside phosphorylase (Pnp1) and pyrimidine uridine hydrolase (Urh1) have been shown to catalyze these reactions (Lecoq *et al*, 2001; Kurtz *et al*, 2002). To investigate if Pnp1 and Urh1 are required for ribose salvage, we examined strains deleted for *PNP1* or *URH1* or both. Following carbon starvation, the *pnp1* strain accumulated more of the nucleosides, inosine and guanosine and less of the

bases hypoxanthine and guanine than did wild-type cells; similarly *urh1* accumulated more uridine and cytidine and less uracil (Figure 5). Adenosine did not accumulate in the *pnp1* strain, consistent with the previous finding that Pnp1 exhibits no activity with adenosine. In addition, guanine, cytosine and cytidine concentrations were markedly lower than xanthine, uracil and uridine, presumably due to the activities of guanine, cytosine and cytidine deaminases Gud1, Fcy1 and Cdd1 respectively (Supplementary Figure S2A). Strains with individual deletions of *PNP1* or *URH1* exhibited decreased S7P and SBP levels upon carbon starvation compared to wild type, indicating that both enzymes contribute to ribose salvage. The double deletion strain accumulated all nucleosides and depleted all bases, and exhibited decreased S7P and SBP levels following carbon starvation to an extent similar as the *phm8* deletion. Thus, Pnp1 and Urh1 work in concert to

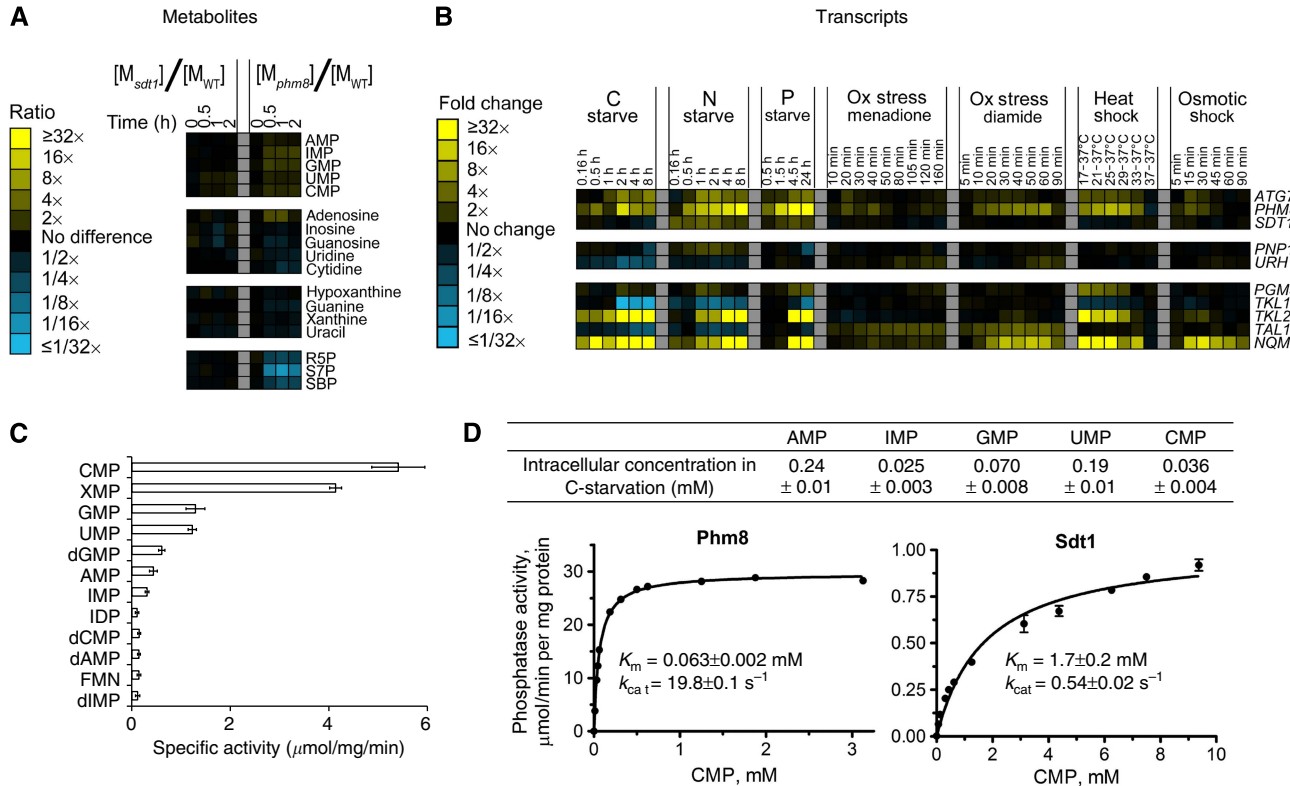

**Figure 4** Phm8 is the physiological yeast nucleotidase. (**A**) Ratio of metabolite levels in *sdt1* and *phm8* strains versus wild-type strain in carbon starvation. All reported values are log$_2$ transformed ratios; data are mean of duplicate samples at each time point. (**B**) Summary of transcripts data of *PHM8*, *SDT1* and other enzymes in the pathway (Gasch *et al*, 2000; Bradley *et al*, 2009; Klosinska *et al*, 2011). All values are log$_2$ transformed fold changes. (**C**) Screening of Phm8's phosphatase activity against 90 phosphorylated compounds. Phosphatase activity was measured in the presence of 0.5 mM substrate and 5 mM Mg$^{2+}$ (pH = 7.0, 30°C). Compounds with specific activity higher than 0.1 μmol/mg/min are shown. The *x* axis represents specific phosphatase activity (μmol of phosphate produced per minute per mg of enzyme, mean ± range of *N* = 2 replicates). (**D**) Top table: absolute intracellular concentration of nucleotide monophosphates in carbon starvation. Bottom plots: Phosphatase activity of Phm8 and Sdt1 as a function of CMP concentration. The *x* axis represents CMP concentration and the *y* axis represents specific phosphatase activity (μmol of phosphate produced per minute per mg of enzyme, mean ± range of *N* = 2 replicates).

convert purine and pyrimidine nucleosides into bases and ribose or ribose-1-phosphate.

Further metabolism of ribose-1-phosphate requires its conversion to ribose-5-phosphate. Phosphoglucomutase 3 (Pgm3) has been proposed to be a phosphoribomutase capable of catalyzing this conversion (Walther *et al*, 2012). Consistent with this proposal, we found that deletion of *PGM3*, but not *PGM1* or *PGM2*, led to accumulation of ribose-1-phosphate and upstream purines in the ribose salvage pathway (Figure 5). Thus, Pgm3 is the physiological phosphoribomutase required for ribose salvage from purines. We accordingly refer to the gene by the new name *PRM15*.

The enzyme responsible for the phosphorylation of ribose generated from pyrimidine degradation remains, however, undefined. An open reading frame (Rbk1) has been annotated as a ribokinase due to its similarity to ribokinases in other organisms. While *S. cerevisiae* cannot utilize ribose as the sole carbon source, in the absence of glucose, it did assimilate extracellular ribose, albeiet at a slow rate (Supplementary Figure S2C, D). Rbk1 deletion only slightly slows extracellular ribose assimilation and does not alter S7P or SBP accumulation during carbon starvation (Supplementary Figure S2B–D). Thus, another enzyme presumably phosphorylates the ribose

produced by pyrimidine nucleoside degradation. As no other yeast enzyme exhibits strong homology to known ribokinases, it is possible that an annotated hexokinase might fulfill this role.

## Sedoheptulose species accumulate due to substrate limitation of transaldolase

To confirm that the sedoheptulose species accumulating during carbon starvation derive from ribose phosphate produced from RNA degradation, we deleted both transketolase isozymes (*TKL1/TKL2*), preventing conversion of pentose phosphates into sedoheptulose7-phosphate. Unlike wild-type, the *tkl1/tkl2* strain did not accumulate S7P and SBP upon carbon starvation (Figure 6A and B). Thus, ribose salvage requires a functional nonoxidative PPP.

The nonoxidative PPP should in principle be capable of converting ribose phosphate entirely into glycolytic intermediates. Thus, the cause of S7P and SBP accumulation under carbon starvation conditions is unclear. In the canonical nonoxidative PPP, S7P is synthesized from two pentose phosphates by transketolase, and then converted by

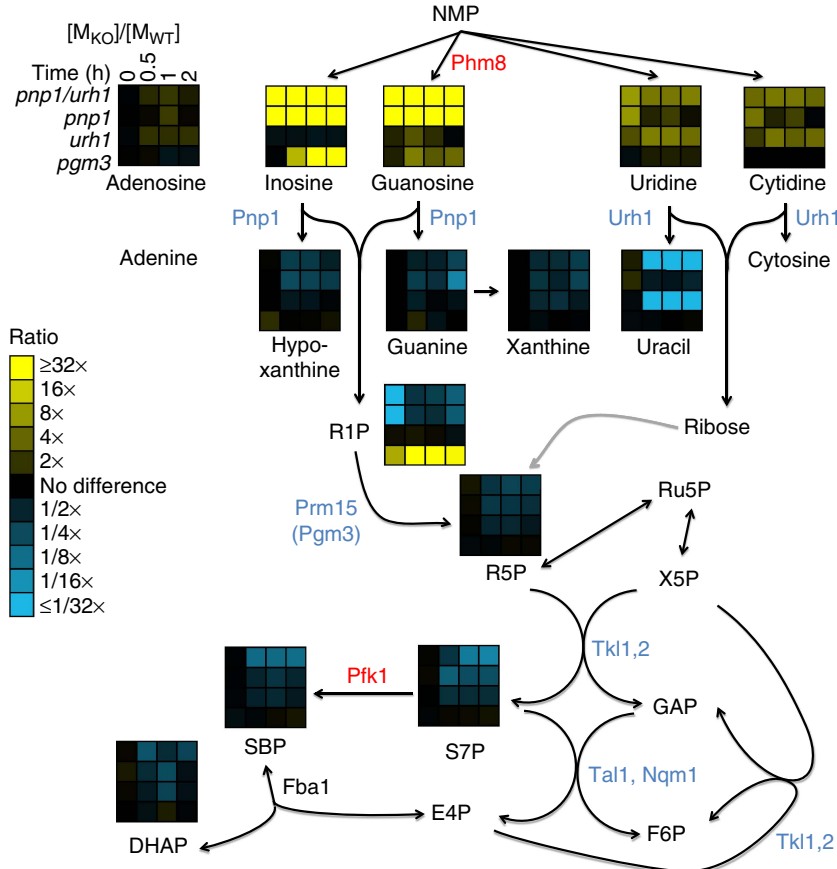

**Figure 5** Confirmation that Pnp1 is the physiological purine nucleoside phosphorylase, Urh1 is the pyrimidine hydrolase, and Prm15 (Pgm3) is the phosphoribomutase. Ratio of metabolite levels in *pnp1/urh1*, *pnp1*, *urh1* and *prm15* (*pgm3*) strains versus wild-type strain during carbon starvation. Data are shown in heat map format, with each line reflecting the dynamics of the ratio of the metabolite levels in a particular strain versus wild-type strain. All reported values are $\log_2$ transformed ratios; data are mean of duplicate samples at each time point.

transaldolase by condensation with glyceraldehyde-3-phosphate (GAP) into F6P and E4P. Indeed, genetic inactivation of both transaldolase isozymes (*tal1/nqm1*) led to chronic S7P accumulation (Figure 6C). Accordingly, depletion of GAP during carbon starvation might prevent S7P consumption by transaldolase and thus lead to S7P accumulation. Although we could not directly measure GAP due to its low abundance, it is in rapid exchange with DHAP, which decreased more than 20-fold in carbon starvation.

To explore whether GAP depletion underlies the S7P accumulation, in lieu of complete carbon starvation, we transferred cells from minimal media containing glucose as the sole carbon source to minimal media containing dihydroxyacetone (DHA) as sole carbon source. DHA can be transported into yeast cells and phosphorylated into DHAP, which then forms GAP. These reactions can support yeast growth, but only after several days of adaptation (Molin *et al*, 2003). In the shorter time scale of our experiments, we found that the switch to DHA produces the same metabolite concentration changes as overt starvation, with only three exceptions: DHAP was two-fold higher and S7P and SBP decreased rather than increased (Figure 6D). Thus, during carbon starvation, GAP depletion precludes S7P catabolism by transaldolase and consequently results in S7P accumulation.

The results from the above experiments present a conundrum regarding the source of SBP. Previous studies have shown that SBP is produced in yeast via a reversible reaction catalyzed by FBP aldolase, whose substrates are DHAP + E4P (Clasquin *et al*, 2011). However, DHAP decreases substantially under carbon starvation. Moreover, as evident from Figures 2, 4, and 5, S7P and SBP concentrations are highly correlated. We also observed that, after feeding yeast with different labeled forms of glucose, S7P and SBP always exhibited identical labeling patterns during subsequent carbon starvation (Supplementary Figure S3). Accordingly, we explored the possibility that SBP can also be formed from S7P.

*E. coli* phosphofructokinase can phosphorylate S7P to SBP (Nakahigashi *et al*, 2009). To test whether this can occur in yeast, we created strains lacking either phosphofructokinases isozyme, Pfk1 or Pfk2, and measured metabolites following carbon starvation. The *pfk1* but not *pfk2* deletion strain had lower SBP and higher S7P (Figure 6E and Supplementary Figure S4), indicating that Pfk1 phosphorylates S7P to SBP, perhaps selectively during carbon starvation when the S7P/F6P ratio (reflected by S7P/G6P ratio, as G6P and F6P are linked via a single reversible reaction) greatly increases (Figure 2). As the *pfk1* strain exhibited lower DHAP compared to the wild-type strain upon carbon starvation, SBP is

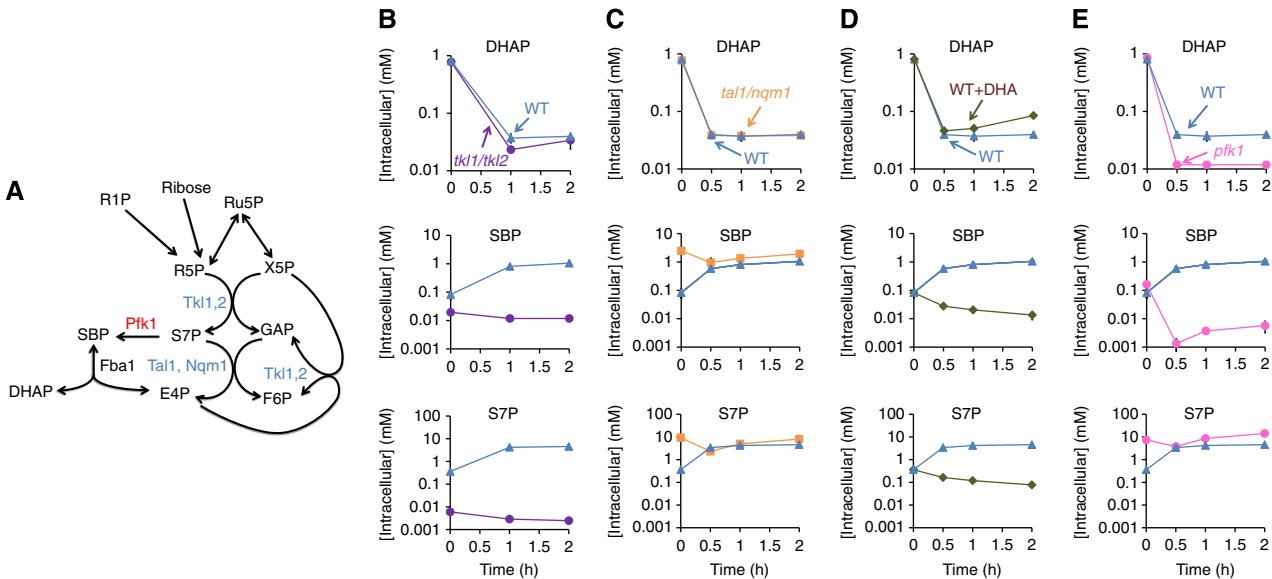

**Figure 6** Nucleoside degradation causes S7P accumulation due to depletion of the other transaldolase substrate glyceraldehyde-3-phosphate, and SBP accumulation due to phosphorylation of S7P into SBP by Pfk1. (**A**) Diagram of proposed reactions. (**B–E**) DHAP, SBP and S7P levels in wild type, *tkl1/tkl2* (**B**), *tal1/nqm1* (**C**) and *pfk1* (**E**) strains upon carbon starvation, and wild-type strain upon abruptly switching from glucose to no carbon (WT) versus to dihydroxyacetone (WT + DHA) as the sole carbon source (**D**). The *x* axis represents hours after carbon starvation, and the logarithmic *y* axis represents absolute intracellular concentration (mean ± range of $N = 2$ biological replicates).

presumably degraded by aldolase into DHAP, providing an alternative route into glycolysis.

## Ribose salvage is essential for yeast's survival in starvation and oxidative stress

The identification of the enzymes of ribose salvage provided us the opportunity to investigate the pathway's physiological role. Ribose salvage could serve to maintain energy charge and levels of key metabolites during nutrient starvation. We confirmed this hypothesis by metabolomic analysis of the ribose salvage mutants *phm8* and *pnp1/urh1* (Figure 7A). Specifically, there was a marked impairment in energy charge in carbon starvation; glutamine abundance (a hallmark of nitrogen availability, Crespo *et al*, 2002; Zaman *et al*, 2008) in nitrogen starvation, and ATP abundance (a hallmark of phosphate availability, Boer *et al*, 2010) in phosphate starvation. These metabolite concentration changes were also associated with altered cell growth and viability: While the *phm8* and *pnp1/urh1* deletion strains grew identically to wild type on complete minimal media, they were less effective than wild type at sustaining growth when nitrogen or phosphorous was removed (Figure 7B). All strains ceased growth immediately upon carbon starvation. However, *phm8* and *pnp1/urh1* strains exhibited substantially slower initiation of gluconeogenesis if switched from glucose to glycerol + ethanol (Figure 7D), presumably due to the inferior carbon supply in the interval between fermentative and respiratory growth. The *phm8* and *pnp1/urh1* strains also exhibited decreased viability in long-term starvation (Figure 7C). These findings indicate that nucleotide degradation and ribose salvage are essential for yeast's growth and survival under nutrient deprivation.

The above results provide a clear physiological role for the ribose salvage pathway. They do not, however, clarify the

function of the S7P accumulation during carbon starvation. Motivated by our observation that S7P can be consumed if GAP is available, we looked for a stress condition under which DHAP and GAP might accumulate. Since oxidative stress inhibits GAP dehydrogenase in yeast (Shenton and Grant, 2003), carbon stored as S7P could be released by a transient increase in GAP triggered by oxidative stress. Moreover, conversion of S7P to F6P could in turn lead to production of glucose-6-phosphate by phosphoglucose isomerase and NADPH production by the oxidative PPP. Subsequent consumption of NADPH to regenerate reduced glutathione could promote survival of oxidative stress. To test this possibility, we added $H_2O_2$ to carbon-starved wild type, *phm8* and *pnp1/urh1* yeast cells (Figure 8A and B). Indeed, DHAP, which is in rapid exchange with GAP, accumulated immediately upon oxidative stress, mirrored by a rapid decrease in S7P and rise in G6P in the wild-type cells (Figure 8C). The *phm8* and *pnp1/urh1* deletion strains, in contrast, exhibited lower G6P, NADPH/NADP$^+$ ratio, reduced glutathione, and viability (Figure 8D and E). Thus, S7P accumulated during short-term carbon starvation enables rapid NADPH production in response to oxidative stress.

## Discussion

Our characterization of nucleotide degradation in yeast represents the most comprehensive metabolomic investigation to date, in any organism, of the pathway from nucleotide monophosphates to central metabolism. Such investigation has substantially clarified the enzymatic steps of the pathway. Because degradative pathways, unlike biosynthetic pathways, are inessential for growth, they cannot be readily mapped by purely genetic approaches relying on overt phenotypes such as dependence on a particular added nutrient (auxotrophy).

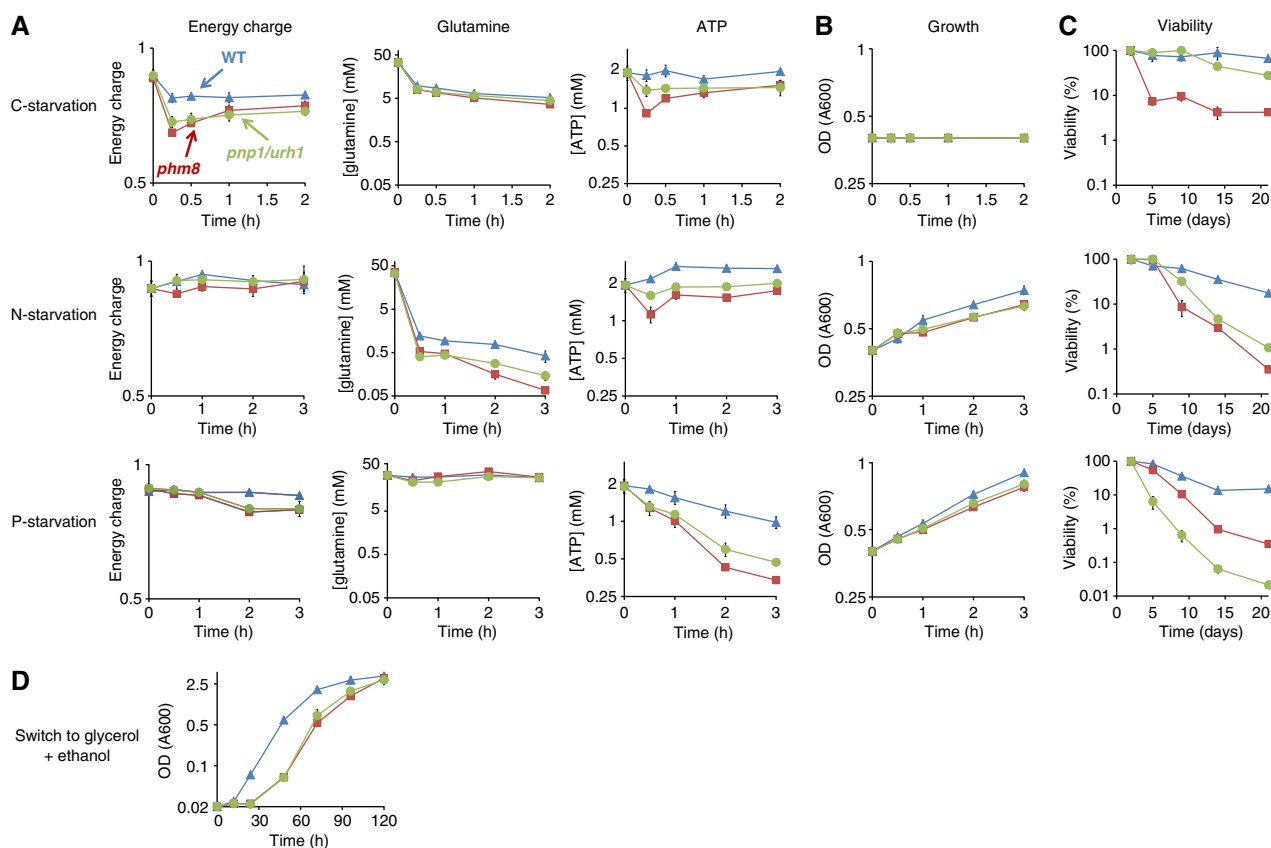

**Figure 7** The ribose-salvage pathway is essential for yeast's survival in starvation. (**A**) Level of hallmark metabolites and energy charge in wild type, *phm8* and *pnp1/urh1* strains in starvation. The *x* axis represents hours after starvation, and the logarithmic *y* axis represents energy charge ([ATP] + 0.5[ADP]/([ATP] + [ADP] + [AMP]) or absolute intracellular concentration (mean ± range of $N = 2$ biological replicates). (**B** and **C**) Growth and viability of wild type, *phm8* and *pnp1/urh1* strains in starvation. (**D**) Growth of wild type, *phm8* and *pnp1/urh1* strains in the switch from glucose to glycerol + ethanol. The *x* axis represents hours after starvation and the logarithmic *y* axis represents optical density ($A_{600}$) in (**B** and **D**) and percentage of live cells calculated by dividing the number of colonies formed after starvation by the number of colonies formed before starvation in (**C**) (mean ± range of $N = 2$ biological replicates).

Moreover, many degradation enzymes have promiscuous activity against a diversity of substrates, rendering their functions difficult to determine from biochemistry alone (Saghatelian and Cravatt, 2005b). For example, in *E. coli*, various enzymes have been putatively identified as nucleotidases (Innes *et al*, 2001; Proudfoot *et al*, 2004; Kuznetsova *et al*, 2006) and nucleosidases (Jensen and Nygaard, 1975; Petersen and Moller, 2001; Dandanell *et al*, 2005), but which one(s) are physiologically important remains unknown. Mammalian nucleotide degradation and ribose salvage are yet more complex and less well defined (Tozzi *et al*, 2006).

To analyze the physiological function of nucleotide degradation enzymes in yeast, we employed metabolomic analysis of targeted deletion strains (Fraenkel, 1992; Raamsdonk *et al*, 2001; Fiehn, 2002; Forster *et al*, 2002; Allen *et al*, 2003; Saghatelian and Cravatt, 2005a), and examined the metabolic response to a perturbation that strongly induces the degradation pathway. This allowed us to identify the physiologically relevant enzyme for all but one reaction of the pathway. The significance of this effort is highlighted by its rectifying multiple different misannotations in the current genome-scale yeast metabolic model, including clarifying the lack of Pnp1 activity against adenosine (Figure 5), the specificity of Urh1 to

pyrimidines (Figure 5), the phosphoribomutase activity of Pgm3 (now Prm15) (Figure 5), and most importantly identifying the physiological yeast nucleotidase to be Phm8 (Figure 4), a gene of the haloacid dehalogenase-like hydrolase family. Intriguingly, the strongest metabolic response to *PHM8* deletion occurred in S7P and SBP levels, which are more sensitive to flux through the ribose salvage pathway than are Phm8's direct substrates and products. Measurement of these more distal metabolites was a key benefit of metabolomics over a more targeted analytical approach.

We initiated our study of the ribose salvage pathway to explore an initial observation that, during starvation, mutants defective in nutrient signaling (*bcy1* and *snf1*) exhibited a distinct pattern of PPP metabolites (Figure 3D) (Xu *et al*, 2012b). This suggested that nutrient signaling kinases might regulate metabolic enzymes of the PPP through direct phosphorylation. Similar to our recent observations regarding control of glycolytic enzymes upon glucose removal and re-feeding (Xu *et al*, 2012b), however, we do not find evidence that direct enzyme phosphorylation is controlling PPP metabolite concentrations or fluxes. Rather, the nutrient-signaling kinases serve to coordinate metabolism and cell ribosome contents through regulation of autophagy. The effect

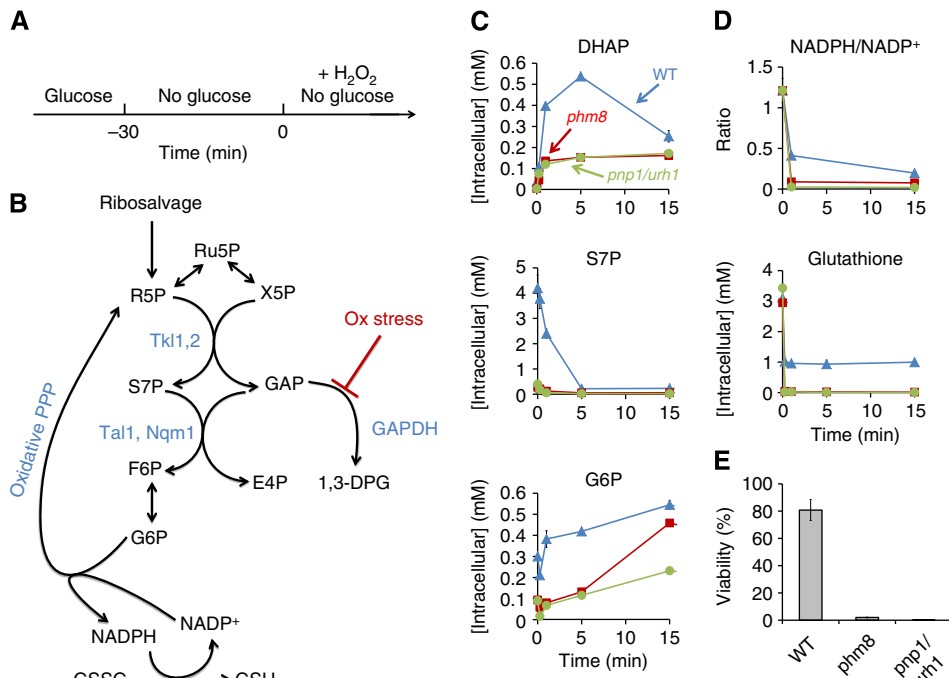

**Figure 8** The accumulation of S7P in carbon starvation facilitates survival of subsequent oxidative stress. (**A**) Experimental design. Yeast cells growing on glucose were switched to no carbon media for 30 min. Thereafter, 20 mM $H_2O_2$ was added and metabolome analyzed by LC-MS. (**B**) Diagram of proposed reactions and their regulation. S7P accumulates during carbon starvation due to low level of glyceraldehyde-3-phosphate (GAP). Oxidative stress blocks GAP consumption via glycolysis by inhibiting GAPDH. This provides sufficient GAP to react with S7P, producing F6P via transaldolase. F6P is further converted to G6P which enters the oxidative PPP. The resulting NADPH is utilized for regeneration of reduced glutathione. 1,3-DPG, 1,3-diphosphoglycerate; GSSG, glutathione disulfide; GSH, reduced glutathione. (**C** and **D**) DHAP, S7P, G6P, and glutathione levels and NADPH/NADP$^+$ ratio in wild type, *phm8* and *pnp1/urh1* strains in the experiment shown in (**A**) The *x* axis represents minutes after oxidative stress, and the *y* axis represents absolute intracellular concentration or ratio of intracellular concentration (mean ± range of $N=2$ biological replicates). (**E**) Viability of wild type, *phm8* and *pnp1/urh1* strains in oxidative stress. The *y* axis represents percentage of survived cells calculated by dividing the number of colonies formed after oxidative stress by the number of colonies formed under the same starvation condition but without $H_2O_2$ treatment (mean ± range of $N=2$ biological replicates).

of these kinases on the ribose salvage pathway intermediates thus arises from mass action as a consequence of the increased production of nucleotide monophosphates via ribophagy.

Beyond being a marker of nucleotide degradation flux, S7P accumulation during carbon starvation provides a potentially useful nutrient reserve of carbon and phosphate. We find that this reserve helps carbon-starved yeast to respond to an acute oxidative stressor. The initial cause of the S7P accumulation in carbon starvation is depletion of glyceraldehyde-3-phosphate, which, when present, reacts with S7P to produce F6P and E4P. Oxidative stress blocks the glycolytic consumption of GAP (Shenton and Grant, 2003), thereby providing the substrate required to react with S7P. The resulting F6P production in turn drives NADPH production via the oxidative PPP and subsequent glutathione reduction.

The accumulation of S7P during carbon starvation is reminiscent of similar accumulation of PEP upon glucose removal. We have recently shown that the increase in PEP is caused by rapid depletion of fructose-1,6-bisphosphate (FBP), an essential activator of the PEP-consuming enzyme pyruvate kinase. The elevated PEP provides a readily accessible reserve of high-energy phosphate bonds, which facilitate fast uptake of glucose if it becomes re-available (Xu *et al*, 2012a, b). Analogously, S7P accumulation is caused by depletion of

FBP's product glyceraldehyde-3-phosphate, a substrate of the S7P-consuming enzyme transaldolase. Like the elevated PEP, the increased S7P prepares the cell for a possible subsequent event, oxidative stress in this case, a major threat for microbes growing on plants (Baker and Orlandi, 1995). In both cases, intrinsic properties of the metabolic network result, during carbon starvation, in carbonaceous metabolite accumulation to provide energy reserves for quick response to further environmental changes.

Beyond the specific role of the S7P accumulation in stress protection, we find that nucleotide degradation is critical to metabolic homeostasis and thus survival of starving yeast. In previous work, during the transient period following glucose upshift when ATP consumption by hexokinase and phosphofructokinase precedes ATP generation by lower glycolysis, AMP degradation was found to help maintain energy charge (Walther *et al*, 2010). In starvation, the goal is not elimination of AMP, but access to the carbon, nitrogen and phosphorous within it and other nucleotide monophosphates. That such constituents would be fed into central metabolism via nucleotide degradation is not unexpected. The significance of their contribution to starvation survival is nevertheless surprising, given that proteins are more copious and carbohydrate stores more accessible. The strikingly lower energy

charge, altered metabolite levels and poor survival in the *phm8* deletion strain, however, show that nucleotide salvage plays an indispensable role. This can be rationalized based on the abundance of ribosomes, which are needed in large amounts only during rapid growth (Mager and Planta, 1991; Bremer and Dennis, 1996; Scott *et al*, 2010). Thus they may be preferentially used as fuel during the transition from fast growth to slow growth or stationary phase.

Building from careful mapping of the nucleotide degradation pathway, we have been able to demonstrate an important role of the pathway in starvation survival of yeast. Orthologs of Phm8 are found in most fungi and plants, and some bacteria and animals but not humans (http://yeast-phylogroups.princeton.edu/). In organisms where Phm8 orthologs are present, our results may directly facilitate mapping of their nucleotide degradation pathways. In other organisms, the approach laid out here may translate even though the gene product does not. For example, although mammalian cells are not normally subject to acute starvation, autophagy is essential for survival of the neonatal period (Kuma *et al*, 2004; Efeyan *et al*, 2012) and contributes to the growth of Ras-driven tumors (Guo *et al*, 2011; Mathew and White, 2011; Yang *et al*, 2011). The latter observation raises the intriguing possibility that a more complete understanding of nucleotide degradation in humans would reveal new drug targets for treatment of aggressive cancers.

# Materials and methods

## Yeast strains and media

Yeast strains were derived from prototrophic S288C or W303. Prototrophic deletions were created by homologous recombination using the allele amplified by PCR from the synthetic genetic analysis (SGA) deletion set (Tong *et al*, 2001). Double deletion strains were made by sporulation and tetrad dissection. Cells were grown in minimal media comprising 6.7 g/l Difco Yeast nitrogen base without amino acids plus 2% (w/v) glucose. For nitrogen and phosphate starvation, cells were grown in nitrogen or phosphate-limited media (Supplementary Table S2) before nutrient deprivation to deplete the otherwise large intracellular store of these two nutrients.

## Yeast culture conditions and extraction

The metabolome of *Saccharomyces cerevisiae* was characterized as described previously (Xu *et al*, 2012b). Briefly, saturated overnight cultures were diluted 1:30 and grown in liquid media in a shaking flask to A600 of ~0.6. A portion of the cells (3 ml) were filtered onto a 50 mm nylon membrane filter (Millipore, Billerica, MA), which was immediately transferred into −20°C extraction solvent (40:40:20 acetonitrile/methanol/water). For nutrient removal, 100 ml of cell culture at A600 of ~0.6 was poured onto a 100-mm cellulose acetate membrane filter (Sterlitech, Kent, WA) resting on a vacuum filter holder with a 1000-ml funnel (Kimble Chase, Vineland, NJ) and was washed with 100 ml prewarmed (30°C) starvation liquid medium lacking the indicated nutrient. Immediately after the wash media went through, the filter was taken off the holder and the cells were washed into a new flask containing 100 ml prewarmed (30°C) starvation medium. Samples were then taken at the indicated time points after nutrient removal and filtered and quenched as described above.

For RNA extraction and quantitation, yeast cells were grown in the same condition as for metabolomics experiments. Total RNA was extracted using the hot acid phenol method followed by ethanol precipitation (Spellman *et al*, 1998). The resulting sample was dissolved in water and total RNA concentration was measured by a fluorometer (Qubit; Invitrogen, Carlsbad, CA).

For cell viability assay, 200 µl of culture were diluted in a 10-fold series, spread onto YPD agar plates and incubated for 24 h. The percentage of cells that survived under a stress was calculated by dividing the number of colonies formed after a stress exposure by the number of colonies formed under the same growth condition but without the stress exposure.

## Metabolite measurement

Cell extracts were analyzed by reversed phase ion-pairing liquid chromatography (LC) coupled by electrospray ionization (ESI) (negative mode) to a high-resolution, high-accuracy mass spectrometer (Exactive; Thermo Fisher Scientific, Waltham, MA) operated in full scan mode at 1 s scan time, $10^5$ resolution, with compound identities verified by mass and retention time match to authenticated standard (Rabinowitz *et al*, 2010). Isomers are reported separately only where they are fully chromatographically resolved. Absolute intracellular metabolite concentrations in steadily growing *S. cerevisiae* were determined as described previously (Bennett *et al*, 2008). Metabolite concentrations after perturbations were computed based on fold-change in ion counts relative to steadily-growing cells (grown and analyzed in parallel) multiplied by the known absolute concentration in the steadily growing cells, as determined using an isotope ratio-based approach (Bennett *et al*, 2009). Energy charge was calculated as $([ATP]+0.5[ADP])/([ATP]+[ADP]+[AMP])$ (Chapman and Atkinson, 1977).

## Protein purification and enzymatic assays

The genes encoding Phm8 (Yer037w) and Sdt1 (Ygl224c) were amplified by PCR using *S. cerevisiae* genomic DNA. The amplified fragments were cloned into a modified pET15b vector (Novagen, Darmstadt, Germany) and overexpressed in the *E. coli* BL21(DE3) Gold strain (Stratagene, La Jolla, California) as previously described (Kuznetsova *et al*, 2010). The recombinant proteins were purified using metal ion affinity chromatography on nickel affinity resin (Qiagen, Hilden, Germany) to high homogeneity and stored at −80°C. Purified Phm8 and Sdt1 were screened for the presence of phosphatase activity against the general phosphatase substrate *p*-nitrophenyl phosphate (*p*NPP) and 90 phosphorylated metabolites as described previously (Kuznetsova *et al*, 2006). For the determination of kinetic parameters ($K_m$ and $k_{cat}$), phosphatase activity was determined in the presence of 5 mM MgCl$_2$ over the range of substrate concentrations (0.001–9.5 mM), and the kinetic parameters were calculated by nonlinear regression analysis of raw data to fit to the Michaelis–Menten function using the GraphPad Prism Software (GraphPad Software, San Diego, CA).

## Supplementary Information

## Acknowledgements

We thank Pat Gibney, Sanford Silverman, Sean Hackett and David Botstein for helpful discussions. This research was funded by NSF CAREER award MCB-0643859, Joint DOE-AFOSR Award DOE DE-SC0002077—AFOSR FA9550-09-1-0580, NSF grant CBET-0941143, and NIH grants R01 grant CA163591 (to JDR) and RO1 GM076562 (to JRB), with additional support from the Princeton University Center for Quantitative Biology (P50 GM071508) and from the Government of Canada through Genome Canada, Ontario Genomics Institute (2009-OGI-ABC-1405) and Ontario Research Fund (ORF-GL2-01-004).

*Author contributions:* Y-F X and JDR designed experiments, analyzed data and wrote the paper. FA, AAC and JRB contributed to strain constructions. WL, EK, GB and AFY contributed to biochemistry. FL contributed to preliminary experiments. FL, AAC, AFY and JRB supported the work and helped with the manuscript.

## Conflict of interest

The authors declare that they have no conflict of interest.

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
