## [Review Process File · Molecular Systems Biology]

Nucleotide degradation and ribose salvage in yeast

Yi-Fan Xu, Fabien Letisse, Farnaz Absalan, Wenyun Lu, Ekaterina Kuznetsova, Greg Brown, Amy A. Caudy, Alexander F. Yakunin, James R. Broach, Joshua D. Rabinowitz

Corresponding author: Joshua D. Rabinowitz, Princeton University

Review timeline:

Submission date:	13 February 2013
Editorial Decision:	27 March 2013
Revision received:	05 April 2013
Accepted:	08 April 2013

Editor: Maria Polychronidou

Transaction Report:

1st Editorial Decision

27 March 2013

Thank you again for submitting your work to Molecular Systems Biology. We have now heard back from the three referees who accepted to evaluate the study. As you will see, the referees find the topic of your study of potential interest and are supportive. However, they raise a series of concerns and make suggestions for modifications, which we would ask you to carefully address in a revision of the present work. The additional analyses proposed by reviewer #2, while not mandatory, would certainly add to the completeness of the study.

On a more editorial level, I would kindly ask you to include the metabolomics data in the Supplementary Information or to deposit them in the appropriate public database. (Additional information is available in the "Guide for Authors" section in our website <<http://www.nature.com/msb/authors/index.html#a3.5.2>>) Furthermore, I would like to ask you to include the links and accession numbers in the "Data Availability" section of your manuscript.

Referee report:

Reviewer #1 (Remarks to the Author):

This Ms. is about the endogenous nucleotide degradation pathways in yeast in (mainly) carbon starvation: the identification of Phm8 as a major player in mononucleotide dephosphorylation and particularly interesting findings of sedoheptulose-7-P (S7P) as a major accumulated intermediate and its dissipation in part involving sedoheptulose-1,7-P2 (SBP). This strong and interesting, if

overstated, work deserves a careful rewrite, and not just attention to the items below.

1 Introduction. Mention here rather than later the Nakahigashi et al. 2009 discovery of a S7P to SBP pathway allowing *E. coli* transaldolase mutants to grow on xylose, so as to make clear the new work is addressing the opposite reaction to the authors' nice report of SBPase (Clasquin et al. 2012). Put in the Introduction instead of p. 10 the motivation of exploring nucleotide turnover. Clarify whether S7P is already known as an abundant intermediate (p.3) or discovered here. Etc.

2. Metabolic diagrams. Most of the key two-substrate two-product interconversions are wrong in the diagrams (e.g, the transketolase reaction $R5P+X5P$ to $S7P+GAP$ is shown as $R5P+S7P$ to $X5P+GAP$ in Fig. 1 etc.; there are at least 10 such errors including 4 in Fig. 7.

3. Metabolite concentrations. The heat maps used here and previously show factors of fold-change, but here a few metabolites are individually presented as concentrations (in Figs 3B and 5-7). Ideally, there should be a Table for concentrations in the unperturbed cultures for the 21 compounds presented in heat maps (including the ribose-1-P of Fig. 5) in the usual replete growth condition and for nitrogen-limitation and phosphate-limitation employed for growth in the nitrogen- and phosphate-starvation experiments (p.12 explains that the limitation-media were necessary to deplete stores). Apart from their relevance in the present work, some of these nominally familiar compounds have barely if ever even been reported as to level. (If concentration requires more information than is available, then the ratio calculation should be qualified as being, e.g., ratios of ion counts [surely differing in efficiency] but the ion counts be given, as in Xu et al. 2012b.)

4. 13-C labeling. p.4 line 19 says that the increasing proportion of unlabeled metabolites during the starvation shows that they come from degradation of macromolecules (likely RNA): clarify by saying that this conclusion is drawn from the fact that the metabolites at 0 min are 0% unlabeled. (It is impressive how the analysis is sensitive enough even for low level metabolites. Is another conclusion that the pools throughout these periods are, nonetheless, largely from material labeled between -70 min and 0 min?)

5. Use of ribose moiety. Consider a slightly different emphasis: As said, top p. 7, the experiment of Fig. 5A nicely fits the role of Pnp1 and of the newly recognized phosphoribomutase activity of Pgm3, i.e., here in use of ribose-1-P (R1P) from purine nucleosides in carbon starvation. The fate of free ribose from *Urh1* is something else, for *S. cerevisiae* is not known to grow well (if at all) on exogenous pentoses and the showing of (implied uptake and) phosphorylation of exogenous ribose in Supp. Fig. 2C deserves to be in the text of the paper (rather than just emphasizing that a nominal ribokinase "Rbk1" ORF is not used). And comment on the rate of ribose use?

6. The S7P pathway. Mechanisms and function of S7P and SBP accumulations here are complicated and do not need be proven here. (For example, the idea of physiological glyceraldehyde-3-P dehydrogenase inactivation in oxidative stress, appears a stretch. But so what.) The relationship to RNA degradation seems reasonable (but Supp. Fig. 1A should at least give amounts and not just relative level), and to triose-P levels is a clever idea. The finding that transketolase blocks alters the key metabolites, Supp. Fig. 3A, deserves to be in the text together with Fig. 5B on the other blocks. It would be helpful to separate the notions of S7P perturbation (i.e., the dihydroxyacetone and hydrogen peroxide experiments), from those on how SBP might arise. .

7. Other items. (i) In Fig. 1 the phosphatase and phosphorylase reactions should show that Pi is made or used; the phosphofruktokinase reaction be indicated as Pfk1,2 (they are not normally separate isozymes); GAPDH is Tdh1-3; and Tpi1 catalyzes a reversible reaction. Fig. 2 should show S7P to SBP? (ii) p. 4 last para., clarify how the *atg7* strain "showed no increase" in unlabeled species (Fig. 3B appears to show several significant increases. (iii) p.8 l.3 says that DHAP was two-fold higher (in the presence of DHA) but Fig. 5B top does not appear to show the unsupplemented WT line? (iv) p.8 l. 9 says DHAP and E4P decrease substantially but where is E4P reported? (Not in Fig. 2.) Is it measurable? What about free ribose (in the normal experiment, not in ribose feeding), and adenine and cytosine (below detection?). (v) p.8 l.18, F6P was not measured, so say S7P/G6P(F6P)? (vi) Redo Supp. Table 2 to allow direct comparison of the major salts in the three basal media (replete, N-limited and P-limited). (vii) Omit Table 3, the list of compounds tried with Phm8 (if it's the same list referred to in Kuznetsova and there are no data other than Fig. 4). (viii) Fig. 5 legend and Supp. Table 1 has "Prm1," for GPM3, a mnemonic already in use (ix) p.12 l. 4

give a reference to the SGA prototrophic deletion set (this is not kan-r nulls?). (x) p.9 para 2 l.3, concludes SBP "consumption" after GAP replenishment. But the experiment appears to show lack of accumulation. (xi) Give concentration of ribose for Supp. Fig. 2C-D. (xii) Supp. Fig. 4 does not appear to discriminate between (the stated) growth lag and slower growth ((xii) Say something about the interesting difference from work of the Toulouse group on adenine nucleotide perturbations in carbon excess. (xiii) Choice of references is unbalanced with, e.g. three on p. 2 to ribosomes having about equal RNA and protein. (xiv) Metabolite levels is hardly a new field, so reconsider words like paradoxical and counterintuitive.

Reviewer #2 (Remarks to the Author):

In the paper Xu et al. present a thorough investigation of the metabolic pathways and enzymes involved in nucleotide salvage in yeast. The paper is interesting and reads very well; the manuscript is important because, as authors suggest, this is probably the best experimental analysis of nucleotide/ribose salvage pathway in any organisms. The authors use a combination of genetics and metabolomics to investigate the relevant changes. The newly identified nucleosidase Phm8, as well as Pnp1/Urh1, are found to be important for metabolic homeostasis following starvation. Also, the authors finding of sedoheptulose-7-phosphate (SS7P) accumulation suggests an interesting role of nucleotide degradation in protection against oxidative stress. Besides minor corrections and comments, I suggest to publish the manuscript in MSB essentially in its current form.

Minor comments:

i On page 4, 3rd paragraph, the authors indicate: " Moreover, the deletion strain showed no increase in unlabeled species when transient feeding of ^{13}C -labeled glucose was followed by carbon starvation (Figure 3B, open symbols)". While the accumulation in the mutant is always smaller than observed for WT, the figure does show an increase in the unlabeled fraction for at least some of the relevant compound. This sentence should be re-written to more faithfully describe the result.

ī For clarity, I suggest in Figure 5 authors label Rrm1 as "pgm3" or "Rrm1 (pgm3)".

ī For general reader, I suggest that the authors explain (and probably give a formula in methods) for the energy charge.

Also to add more systems biology flavor to the paper, I suggest two analyses that the authors can do relatively quickly.

- There is a large database of expression data for yeast. I suggest that authors analyze the expression of Phm8 in multiple growth and stress conditions. And maybe also expression of other relevant enzymes from the pathway. I think the results will be interesting, especially in light of the possible role in oxidative stress relief. The authors may add an expression figure to the manuscript; something like expression of relevant enzymes versus phenotypes. It will be also interesting to understand the enzymes and regulatory genes with high Phm8 co-expression.

- It will be interesting to understand the phylogenetic distribution of Phm8 orthologs in other organism (microbes), or at least in related yeast species. This is especially interesting as the nucleotide salvage pathway(s) are not mapped well across species.

Reviewer #3 (Remarks to the Author):

This highly interesting manuscript describes a previously unknown role for sedoheptulose phosphates in the mobilisation of carbon from ribosomes during yeast autophagy triggered by carbon starvation. Convincing evidence is presented supported by metabolomics measurements and isotopic flux analysis. In addition, analysis of deletion mutants and enzyme kinetic measurements points to the need for re-annotation of some yeast genes. There is a realistic chance that this system might operate in other organisms during starvation or senescence, adding additional importance to the discovery.

I have no major criticisms of the scientific content of the paper.

A minor presentational criticism is that the representation of the pentose phosphate pathway reactions in diagrams 1, 2, 5 and 7 is confusing even to an old-fashioned biochemist such as myself who thought he knew the pathway. The two main reasons for this are firstly that the two reactions

from R5P to X5P are omitted throughout, even though they must carry as much flux as the R5P consumption by transketolase, and the current presentation gives the impression that X5P is being recycled within the pathway. Secondly, the double reaction arrows for transaldolase and transketolase reactions do not follow normal biochemical convention; they need rotating through 90 degrees. (Though they correctly imply these reactions are reversible, it might even be clearer if they were shown in this context as unidirectional as required for R5P consumption.)

Finally, a suggestion for consideration by the authors: is it possible that another function of this system is to retain the phosphate component of the RNA in soluble organic form in the cytosol? Since 7 R5Ps are converted to 5 SBP/S7P, the phosphate equivalent of the ribose can be stored.

1st Revision - authors' response

05 April 2013

Substantive Changes

Reviewers raised three primary concerns:

- 1. The expression pattern of Phm8 and relevant enzymes in the pathway.** To address this issue, we explored ~300 transcripts datasets available for yeast and selected conditions where Phm8 is highly expressed. These conditions included carbon, nitrogen and phosphate starvation, oxidative stress, osmotic stress and heat shock. This is consistent with Phm8's role in stress conditions where the degradation of RNA might be essential for yeast's survival. The expression of Phm8 is also correlated with Atg7, Pgm3 and Tkl2 in the pathway, indicating that these enzymes work in concert to degrade RNA and salvage its components. We added, as new Figure 4A, a summary heat map for the transcripts data.
- 2. The pentose phosphate pathway arrows in the figures are confusing.** We changed bi-directional arrows into unidirectional ones for transaldolase and transketolases. We also rotated the arrows 90 degree to better illustrate the combination of substrates and products. Moreover, we added inorganic phosphates in Figure 1 to make all metabolic reactions balanced.
- 3. Suppl. Figure 4 didn't discriminate between (the stated) growth lag and slower growth.** The previous experiment was done at a high OD (starting at 0.3) so there was not enough time to fully discriminate the growth lag and slower growth. To address this issue, we repeated the same switch experiment (glucose → glycerol + acetate) at a much lower OD. The new data (now Figure 7D) clearly indicates that *phm8* and *pnp1/urh1* strains are defective in the growth lag phase.

Point-by-point response to reviewers:

Reviewer #1 (Remarks to the Author):

This Ms. is about the endogenous nucleotide degradation pathways in yeast in (mainly) carbon starvation: the identification of Phm8 as a major player in mononucleotide dephosphorylation and particularly interesting findings of sedoheptulose-7-P (S7P) as a major accumulated intermediate and its dissipation in part involving sedoheptulose-1,7-P2 (SBP). This strong and interesting, if overstated, work deserves a careful rewrite, and not just attention to the items below.

We appreciate the reviewer's specific suggestions and also the concern that the initial paper may have been overstated in certain cases. We have acted on the suggestions and also carefully checked the paper to minimize possible overstatements. However, as the other reviewers found the paper overall well written, we did not consider it productive to rewrite most of the paper.

1 Introduction. Mention here rather than later the Nakahigashi et al. 2009 discovery of a S7P to SBP pathway allowing E. coli transaldolase mutants to grow on xylose, so as to make clear the new work

is addressing the opposite reaction to the authors' nice report of SBPase (Clasquin et al. 2012). Put in the Introduction instead of p. 10 the motivation of exploring nucleotide turnover. Clarify whether S7P is already known as an abundant intermediate (p.3) or discovered here. Etc.

We agree with the reviewer that the Nakahigashi paper is a better reference in the introduction since we are focusing on the direction from pentose phosphate pathway to glycolysis. Also, in the Nakahigashi paper, the S7P → SBP reaction was found in the transaldolase deletion strain where our study discovered this reaction in wild type cells under stress conditions. We have included these information in the introduction.

We also included the absolute concentration of S7P determined by isotope ratio method. Moreover, we added a supplementary table to include the absolute concentration of all the metabolites monitored in this study (see response to comment 3).

2. Metabolic diagrams. Most of the key two-substrate two-product interconversions are wrong in the diagrams (e.g, the transketolase reaction $R5P+X5P \rightarrow S7P+GAP$ is shown as $R5P+S7P \rightarrow X5P+GAP$ in Fig. 1 etc.; there are at least 10 such errors including 4 in Fig. 7.

While different papers in the literature followed different conventions for PPP arrows and the initial submission was therefore not incorrect, we concur that the more common and clear presentation is different from the initial submission. We thank the reviewer for pointing this out and have corrected the paper accordingly.

3. Metabolite concentrations. The heat maps used here and previously show factors of fold-change, but here a few metabolites are individually presented as concentrations (in Figs 3B and 5-7). Ideally, there should be a Table for concentrations in the unperturbed cultures for the 21 compounds presented in heat maps (including the ribose-1-P of Fig. 5) in the usual replete growth condition and for nitrogen-limitation and phosphate-limitation employed for growth in the nitrogen- and phosphate-starvation experiments (p.12 explains that the limitation-media were necessary to deplete stores). Apart from their relevance in the present work, some of these nominally familiar compounds have barely if ever even been reported as to level. (If concentration requires more information than is available, then the ratio calculation should be qualified as being, e.g., ratios of ion counts [surely differing in efficiency] but the ion counts be given, as in Xu et al. 2012b.)

We agree with the reviewer that absolute concentration of the metabolites monitored in this study is very important for better understanding the dynamic change of metabolism. So we have measured the concentration of these metabolites using the internally isotope ratio method described previously in our lab (Bennett Nat. Chem. Biol. 2009) and made a supplementary table (Table S3) containing these important data. For sugar phosphates, we measured their concentration on replete growth condition (time = 0 in C-starvation). For bases and nucleosides, we measured their absolute total concentration in C-starvation (1 hour) condition because levels of these metabolites are too low to be accurately determined in replete condition. To help facilitate exporting our data for readers, we included another supplementary dataset containing all the fold change data shown in the heat maps.

4. 13-C labeling. p.4 line 19 says that the increasing proportion of unlabeled metabolites during the starvation shows that they come from degradation of macromolecules (likely RNA): clarify by saying that this conclusion is drawn from the fact that the metabolites at 0 min are 0% unlabeled. (It is impressive how the analysis is sensitive enough even for low level metabolites. Is another conclusion that the pools throughout these periods are, nonetheless, largely from material labeled between -70 min and 0 min?)

We realized that this was a mistake we made. For the low abundant metabolites (nucleosides and bases, the first two rows of Figure 3B) at time zero, instead of zero unlabeled, the labeling patterns of these metabolites were actually not quantifiable. We mistakenly calculated them as zero because we didn't detect the unlabeled form. By reviewing our raw data, we found that the labeled forms are also merely at the detection limit. So we have deleted the time zero data for those low abundant metabolites. For high abundant ones (R5P, S7P and SBP), we have double checked the data and they were correct.

5. Use of ribose moiety. Consider a slightly different emphasis: As said, top p. 7, the experiment of Fig. 5A nicely fits the role of Pnp1 and of the newly recognized phosphoribomutase activity of

Pgm3, i.e., here in use of ribose-1-P (R1P) from purine nucleosides in carbon starvation. The fate of free ribose from Urh1 is something else, for *S. cerevisiae* is not known to grow well (if at all) on exogenous pentoses and the showing of (implied uptake and) phosphorylation of exogenous ribose in Supp. Fig. 2C deserves to be in the text of the paper (rather than just emphasizing that a nominal ribokinase "Rbk1" ORF is not used). And comment on the rate of ribose use?

We agree with the reviewer that the utilization of ribose is an important issue here. We repeated many conditions and found that S. cerevisiae cannot use ribose as the sole carbon source. Although there is some assimilation of ribose in the absence of glucose, it is apparently too slow to support growth. Consistent with this, R5P got labeled very slowly (comparing to the rapid and almost complete labeling of G6P when feeding glucose). We now discussed the rate of ribose use in the Rbk1 section in the main text.

6. The S7P pathway. Mechanisms and function of S7P and SBP accumulations here are complicated and do not need be proven here. (For example, the idea of physiological glyceraldehyde-3-P dehydrogenase inactivation in oxidative stress, appears a stretch. But so what.) The relationship to RNA degradation seems reasonable (but Supp. Fig. 1A should at least give amounts and not just relative level), and to triose-P levels is a clever idea. The finding that transketolase blocks alters the key metabolites, Supp. Fig. 3A, deserves to be in the text together with Fig. 5B on the other blocks. It would be helpful to separate the notions of S7P perturbation (i.e., the dihydroxyacetone and hydrogen peroxide experiments), from those on how SBP might arise.

We agree with the reviewer that showing absolute amount or moles of ribose in RNA is more helpful to connect the decrease of RNA to the accumulation of degradation products. We have modified the units in Supp. Fig. 1A to show the mM values of ribose in RNA. This is because all metabolite concentrations used in the paper are in mM.

We have put the transketolase data in the main text. We also separate the original Figure 5 into two figures. The current Figure 5 contains only the steps from nucleosides to ribose moieties, while the current Figure 6 contains the data for non-oxidative PPP mutants and +DHA experiment. We also separate the current Figure 6 into several panels for clarity.

7. Other items. (i) In Fig. 1 the phosphatase and phosphorylase reactions should show that Pi is made or used; the phosphofructokinase reaction be indicated as Pfk1,2 (they are not normally separate isozymes); GAPDH is Tdh1-3; and Tpi1 catalyzes a reversible reaction. Fig. 2 should show S7P to SBP?

We have added Pi as requested.

While Pfk1 and Pfk2 may form a protein complex, each has independent catalytic activity. We find that only the deletion of Pfk1, but not Pfk2, increases the concentration of S7P and decreases the concentration of SBP in C-starvation. So we concluded that Pfk1 is the main enzyme converting S7P to SBP. We have included a new supplementary figure (now figure S4) for the Pfk2 data.

(ii) p. 4 last para., clarify how the atg7 strain "showed no increase" in unlabeled species (Fig. 3B appears to show several significant increases).

We agree with the reviewer than "showed no increase" is a too strong argument. We have softened this argument to "showed less increase"

(iii) p.8 l.3 says that DHAP was two-fold higher (in the presence of DHA) but Fig. 5B top does not appear to show the unsupplemented WT line?

We are sorry that the WT line was hidden under the tall line. We have made it visible.

(iv) p.8 l. 9 says DHAP and E4P decrease substantially but where is E4P reported? (Not in Fig. 2.) Is it measurable? What about free ribose (in the normal experiment, not in ribose feeding), and adenine and cytosine (below detection?).

Indeed E4P has very low abundance. Although we tried many times, we didn't get publishable data on this compound. So we have deleted the argument about the E4P and kept only DHAP. This doesn't affect our original argument since the decrease of DHAP and no detectably increase of E4P

indicates that the reaction cannot run from DHAP + E4P -> SBP. For ribose, it is not quantifiable from our mass spec set up due to its low ionization efficiency and abundance. Same detection problem also applied to adenine and cytosine whose concentration is below 1 μ M in the cell.

(v) p.8 l.18, F6P was not measured, so say S7P/G6P(F6P)?

We have made the requested modification.

(vi) Redo Supp. Table 2 to allow direct comparison of the major salts in the three basal media (replete, N-limited and P-limited).

We have re-made the table to show the comparison of the salts content in our media. We agree that this is a better presentation of the media composition. Thanks for the suggestion.

(vii) Omit Table 3, the list of compounds tried with Phm8 (if it's the same list referred to in Kuznetsova and there are no data other than Fig. 4).

Agree. We have deleted the table.

(viii) Fig. 5 legend and Supp. Table 1 has "Prm1," for GPM3, a mnemonic already in use

We have changed all the Prm1 into Prm15. Thanks for reminding us Prm1 is already in use.

(ix) p.12 l. 4 give a reference to the SGA prototrophic deletion set (this is not kan-r nulls?).

The prototrophic deletion strains were created from the synthetic genetic analysis (SGA) deletion set (BY4743 background, Kan-r nulls) from the study "Systematic genetic analysis with ordered arrays of yeast deletion mutants" in Tong 2001 Science paper. We have included this reference in the methods section.

(x) p.9 para 2 l.3, concludes SBP "consumption" after GAP replenishment. But the experiment appears to show lack of accumulation.

Thank you for pointing out that we directly measured DHAP not GAP. DHAP indeed rises upon H₂O₂ addition (now Figure 8C). We have changed the text to refer to DHAP not GAP.

(xi) Give concentration of ribose for Supp. Fig. 2C-D.

We have included the concentration in the figure legend of Supp. Fig. 2C-D.

(xii) Supp. Fig. 4 does not appear to discriminate between (the stated) growth lag and slower growth

We agree with the reviewer that we cannot discriminate between the growth lag and slower growth in the previous Supp. Fig. 4. This might be due to the fact that the switch was performed at OD = 0.3, which is too high for us to monitor enough generations. To address this issue, we repeated the experiment starting at a much lower OD (0.02). The new data clearly indicate that phm8 and prp1/urh1 cells are defective under growth lag but not at their exponential growth phase. We included the data as the new Figure 7D. Thank the reviewer for the careful inspection, which fundamentally improves this figure.

((xii) Say something about the interesting difference from work of the Toulouse group on adenine nucleotide perturbations in carbon excess.

We agree with the reviewer that the Walther 2010 MSB study is related to ours and the difference deserves to be discussed in the main text. So we have included the comparison of our study with the Walther 2010 study in the discussion.

(xiii) Choice of references is unbalanced with, e.g., three on p. 2 to ribosomes having about equal RNA and protein.

We agree with the reviewer that we don't need three references here. We have deleted the 60S and 40S ones and only left the Ben-Shem 2011 paper on the 80S. We also carefully checked other references to make sure they are appropriately balanced.

(xiv) Metabolite levels is hardly a new field, so reconsider words like paradoxical and counterintuitive.

Thanks for the suggestion. We have changed the words accordingly. We also reviewed the entire manuscript and made changes to wording to make it more readable for general readers.

Reviewer #2 (Remarks to the Author):

In the paper Xu et al. present a thorough investigation of the metabolic pathways and enzymes involved in nucleotide salvage in yeast. The paper is interesting and reads very well; the manuscript is important because, as authors suggest, this is probably the best experimental analysis of nucleotide/ribose salvage pathway in any organisms. The authors use a combination of genetics and metabolomics to investigate the relevant changes. The newly identified nucleosidase Phm8, as well as Pnp1/Urh1, are found to be important for metabolic homeostasis following starvation. Also, the authors finding of sedoheptulose-7-phosphate (SS7P) accumulation suggests an interesting role of nucleotide degradation in protection against oxidative stress. Besides minor corrections and comments, I suggest to publish the manuscript in MSB essentially in its current form.
Minor comments:

- On page 4, 3rd paragraph, the authors indicate: " Moreover, the deletion strain showed no increase in unlabeled species when transient feeding of ^{13}C -labeled glucose was followed by carbon starvation (Figure 3B, open symbols)". While the accumulation in the mutant is always smaller than observed for WT, the figure does show an increase in the unlabeled fraction for at least some of the relevant compound. This sentence should be re-written to more faithfully describe the result.

We agree with the reviewer that we should soften our argument here. We have changed the sentence into: "Moreover, the deletion strain showed less increase in unlabeled species when transient feeding of ^{13}C -labeled glucose was followed by carbon starvation". Thanks for the correction.

- For clarity, I suggest in Figure 5 authors label Rrm1 as "pgm3" or "Rrm1 (pgm3)".

Since Prm1 has already been annotated to another ORF, Ynl279w, we have changed all Prm1 into Prm15. In addition, in most places in the figures we labeled as Prm15 (Pgm3) for clarity. Thanks for the suggestion.

- For general reader, I suggest that the authors explain (and probably give a formula in methods) for the energy charge.

Done.

Also to add more systems biology flavor to the paper, I suggest two analyses that the authors can do relatively quickly.

- There is a large database of expression data for yeast. I suggest that authors analyze the expression of Phm8 in multiple growth and stress conditions. And maybe also expression of other relevant enzymes from the pathway. I think the results will be interesting, especially in light of the possible role in oxidative stress relief. The authors may add an expression figure to the manuscript; something like expression of relevant enzymes versus phenotypes. It will be also interesting to understand the enzymes and regulatory genes with high Phm8 co-expression.

Upon exploring over 300 datasets via Spell, we found that Phm8's expression is induced in various stresses, including starvation, oxidative stress, osmotic stress and heat shock, and is highly

correlated with Atg7. This is consistent with their role of degrading RNA in stress conditions. Although the expression pattern of nucleosidases, Urh1 and Pnp1 is not similar to that of Phm8, the next downstream steps, Pgm3, Tkl2 are expressed under similar stress conditions. In comparison, Sdt1, the previously annotated nucleotidase, was not expressed in such conditions, confirming that Phm8 serves as the major nucleotidase. We have included a panel summarizing the expression data in Figure 4. Thank the reviewer for the helpful suggestion.

- It will be interesting to understand the phylogenetic distribution of Phm8 orthologs in other organism (microbes), or at least in related yeast species. This is especially interesting as the nucleotide salvage pathway(s) are not mapped well across species.

Upon searching Phm8 orthologs in other organisms, we found that it is highly conserved in fungi and plant species. This is now discussed.

Reviewer #3 (Remarks to the Author):

This highly interesting manuscript describes a previously unknown role for sedoheptulose phosphates in the mobilisation of carbon from ribosomes during yeast autophagy triggered by carbon starvation. Convincing evidence is presented supported by metabolomics measurements and isotopic flux analysis. In addition, analysis of deletion mutants and enzyme kinetic measurements points to the need for re-annotation of some yeast genes. There is a realistic chance that this system might operate in other organisms during starvation or senescence, adding additional importance to the discovery.

I have no major criticisms of the scientific content of the paper.

A minor presentational criticism is that the representation of the pentose phosphate pathway reactions in diagrams 1, 2, 5 and 7 is confusing even to an old-fashioned biochemist such as myself who thought he knew the pathway. The two main reasons for this are firstly that the two reactions from R5P to X5P are omitted throughout, even though they must carry as much flux as the R5P consumption by transketolase, and the current presentation gives the impression that X5P is being recycled within the pathway. Secondly, the double reaction arrows for transaldolase and transketolase reactions do not follow normal biochemical convention; they need rotating through 90 degrees. (Though they correctly imply these reactions are reversible, it might even be clearer if they were shown in this context as unidirectional as required for R5P consumption.)

We agree with the reviewer that the presentations of the arrows and the non-oxidative pentose phosphate pathway were confusing. We have made the following changes: 1. We added ribulose-5-phosphate into Figure 1, 2, 5, 6 and 8 to show that X5P was made from Ru5P. 2. We rotated the arrows of transaldolase and transketolase 90 degrees and only showed the physiologically favorable direction under the starvation conditions relevant to the current paper.

Finally, a suggestion for consideration by the authors: is it possible that another function of this system is to retain the phosphate component of the RNA in soluble organic form in the cytosol? Since 7 R5Ps are converted to 5 SBP/S7P, the phosphate equivalent of the ribose can be stored.

We agree with the reviewer that this is another possible function of the pathway and have added a sentence to the discussion accordingly.